# ECONet: Efficient Convolutional Online Likelihood Network for Scribble-based Interactive Segmentation

**Muhammad Asad**[1]                                                                    MUHAMMAD.ASAD@KCL.AC.UK
**Lucas Fidon**[1]                                                                            LUCAS.FIDON@KCL.AC.UK
**Tom Vercauteren**[1]                                                            TOM.VERCAUTEREN@KCL.AC.UK
[1] *School of Biomedical Engineering & Imaging Sciences, King's College London, UK*

**Editors:** Accepted at MIDL 2022

## Abstract

Automatic segmentation of lung lesions associated with COVID-19 in CT images requires large amount of annotated volumes. Annotations mandate expert knowledge and are time-intensive to obtain through fully manual segmentation methods. Additionally, lung lesions have large inter-patient variations, with some pathologies having similar visual appearance as healthy lung tissues. This poses a challenge when applying existing semi-automatic interactive segmentation techniques for data labelling. To address these challenges, we propose an efficient convolutional neural networks (CNNs) that can be learned online while the annotator provides scribble-based interaction. To accelerate learning from only the samples labelled through user-interactions, a patch-based approach is used for training the network. Moreover, we use weighted cross-entropy loss to address the class imbalance that may result from user-interactions. During online inference, the learned network is applied to the whole input volume using a fully convolutional approach. We compare our proposed method with state-of-the-art using synthetic scribbles and show that it outperforms existing methods on the task of annotating lung lesions associated with COVID-19, achieving 16% higher Dice score while reducing execution time by 3× and requiring 9000 lesser scribbles-based labelled voxels. Due to the online learning aspect, our approach adapts quickly to user input, resulting in high quality segmentation labels. Source code for ECONet is available at: https://github.com/masadcv/ECONet-MONAILabel.

## 1. Introduction

COVID-19 causes pneumonia-like symptoms, adversely affecting respiratory systems in some patients. In their response to the disease, clinicians have used Computed Tomography (CT) imaging to assess the amount of lung damage and disease progression by localizing lung lesions (Roth et al., 2021; Revel et al., 2021; Rubin et al., 2020). This has been essential in providing relevant treatment for COVID-19 patients with severe conditions and has resulted in acquisition of large number of CT volumes from COVID-19 patients (Roth et al., 2021; Tsai et al., 2021; Wang et al., 2020; Revel et al., 2021). Deep learning-based automatic lung lesion segmentation methods may ease burden on clinicians, however, these methods require large amounts of manually labelled data (Wang et al., 2020; Gonzalez et al., 2021; Tilborghs et al., 2020; Chassagnon et al., 2020). Labelling CT volumes for lung lesion is a time-intensive task which requires expert knowledge, putting further strain on clinicians' workload. In addition, future variants of novel coronaviruses may result in variations in lesion pathologies (McLaren et al., 2020). In such cases, automatic segmentation methods that are trained on existing datasets may fail. To address this, rapid labelling of relevant data is needed to augment existing dataset with new labelled volumes.

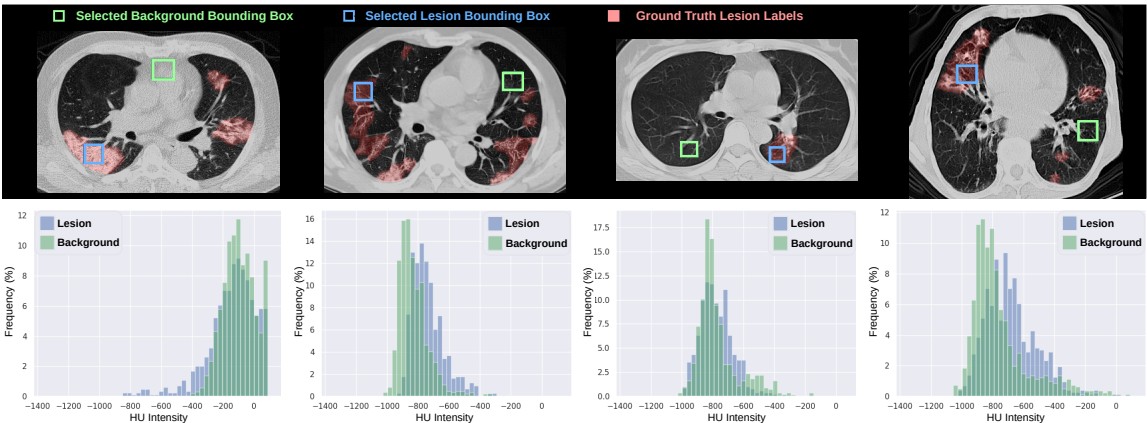

Figure 1: Comparison of appearance of lung lesions vs non-lesions in CT volumes from COVID-19 dataset (Wang et al., 2020). Top row shows a selected slice, and bottom row shows the distribution of HU intensity values for each bounding box. The large overlap between background and lesion distributions indicates that the appearance of some parts of lesion may look similar to background, leading to ambiguity in likelihood models learned from appearance-based features alone.

**Related work.** Due to their quick adaptability and efficiency, a number of existing online likelihood methods have been applied as semi-automatic methods for interactively segmenting objects in images (Boykov and Jolly, 2001; Criminisi et al., 2008; Rother et al., 2004; Barinova et al., 2012; Wang et al., 2016). One of the first approach for interactive segmentation used histogram of intensity values for generating likelihood (Boykov and Jolly, 2001), which was then regularized using a conditional random field formulation solved using a max-flow algorithm. Similarly, (Criminisi et al., 2008) also used histogram-based likelihood for interactively segmenting objects using geodesic symmetric filtering for regularization. In (Rother et al., 2004), a set of Gaussian Mixture Models (GMMs) were employed to model class-specific intensity distribution.

While the intensity-based methods provided significant advancement in terms of interactively segmenting an object, they failed to model ambiguous cases, e.g., where the object intensity is similar to that of the background. To bypass this limitation, hand-crafted features were employed to build online likelihood models in (Barinova et al., 2012; Wang et al., 2016). Barionova et al. (Barinova et al., 2012) proposed an Online Random Forests (ORF) trained using fixed class weights. Dynamically Balanced Online Random Forests (DybaORF) (Wang et al., 2016) utilized dynamically changing weights based on distribution of classes after each user-interaction. Both ORF and DybaORF used hand-crafted features, where DybaORF outperformed all existing online likelihood methods.

Existing online likelihood methods either directly depend on intensity values (Boykov and Jolly, 2001; Criminisi et al., 2008; Rother et al., 2004) or utilize hand-crafted features (Barinova et al., 2012; Wang et al., 2016). While these methods work well for cases where appearance/features for object and background differ sufficiently, they result in failure for cases where this assumption breaks. As shown in Figure 1, the appearance of lung lesions

in COVID-19 patients may have ambiguity, where the distribution of their HU intensity may appear similar to background regions.

A number of deep learning-based interactive segmentation methods exist that provide AI-assisted annotation (Luo et al., 2021; Wang et al., 2018b,a; Rajchl et al., 2016). DeepCut (Rajchl et al., 2016) used bounding box provided by user to train CNNs for fetal brain and lung segmentation from MRI. DeepIGeoS (Wang et al., 2018b) combined CNNs with user-provided scribbles interaction in a two stage CNN approach, where the first stage inferred an initial segmentation and the second refined it using user-scribbles. BIFSeg (Wang et al., 2018a) utilized bounding box interactions with image-specific fine-tuning of CNN to segment unseen objects. MIDeepSeg (Luo et al., 2021) incorporated user-clicks with input image using exponential geodesic distance for interactive segmentation. Deep learning-based interactive segmentation methods consist of large networks that require offline pre-training on large labelled datasets. Additionally, due to the amount of parameters, these networks do not adapt quickly in an online setting to changes in unseen examples. Some methods, such as BIFSeg, propose to use image-specific fine-tuning, however this has limited application in online on-the-fly learning due to extensive computational requirements.

**Contributions.** To address the challenge of learning a distinctive likelihood model in an online and data-light manner, we propose a method which we refer to as Efficient Convolutional Online likelihood Network (ECONet). To the best of our knowledge, ECONet is the first online likelihood method that enables joint and efficient on-the-fly learning of both features and classifier using only scribbles-based labels. The proposed model is lightweight, using only a single convolutional feature layer and three fully-connected layers and can be learned online, while the user provides labels interactively, without the need for any pre-training. We propose an efficient online training technique, where only the patches extracted from scribble-labelled voxels are used. Efficient inference from ECONet is achieved through fully convolutional application of the network on whole input volume (Long et al., 2015). We evaluate ECONet on the problem of labelling lung lesions in CT volumes from COVID-19 patients, with comparison against high-quality segmentation labels from expert annotators. We show that the proposed ECONet outperforms existing state-of-the-art online likelihood methods (Boykov and Jolly, 2001; Rother et al., 2004; Wang et al., 2016), achieving 16% higher Dice score in $3\times$ lower online training and inference time and requiring approximately 9000 lesser interactively labelled voxels.

## 2. Method

### 2.1. Problem Formulation

Let $X = (x_i)_{i=1}^{n} \in \mathbb{R}^n$ represents an image volume that is to be labelled, where $i$ is the index of a given voxel. Given $X$, the user provides scribble-based interaction indicating class labels for a subset of voxels of the image $X$. Let $S = S^f \cup S^b$ represent the set of scribbles, where $S^f$ and $S^b$ denote the foreground and background scribbles, respectively, and $S^f \cap S^b = \emptyset$. For a given voxel $i$, the provided scribble label is $s_i = 1$ if $i \in S^f$ and $s_i = 0$ if $i \in S^b$. The scribbles in $S$ and image patches centered at each scribbles $S$ are used for online training of a given model with parameters $\theta$.

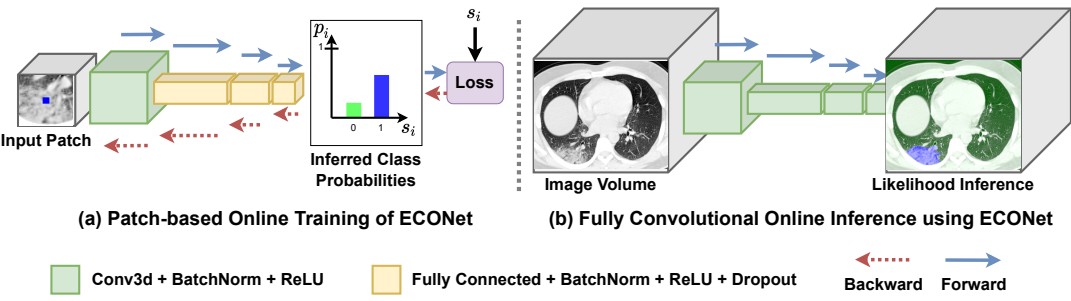

**(a) Patch-based Online Training of ECONet**    **(b) Fully Convolutional Online Inference using ECONet**

Conv3d + BatchNorm + ReLU        Fully Connected + BatchNorm + ReLU + Dropout        Backward    Forward

Figure 2: ECONet online training and inference shows (a) patch-based online training of ECONet where a patch of size K × K × K, extracted around a scribble voxel, is used. The loss function in Eq. (1) is used along with label from scribble to learn the model parameters. (b) shows online likelihood inference using ECONet as fully convolutional network on full image volume.

### 2.2. Online Training and Inference using ECONet

The proposed Efficient Convolutional Online Likelihood Network (ECONet) is a lightweight fully convolutional neural network designed to be trained and applied in an online setting. ECONet consists of one convolution layer used for learning relevant features, which is followed by three fully-connected layers that enable learning the classifier for a given voxel. Each layer is followed by a batch normalization and ReLU activation. To train and apply ECONet in an online setting and using only scribbles-based labels from a user, we propose to use a training and inference strategy that maximizes the efficiency of both tasks. Figure 2 shows an overview of the proposed online training and inference method.

Scribbles $S$ provided by an annotator at a given stage only label a small subset of voxels within a given image volume $X$. Based on this observation, we minimize the computational budget required to perform training passes on ECONet by using K×K×K kernel for input convolution and, extracting and learning only from patches with K×K×K dimensions, each centered around a voxel with user-scribble (Figure 2 (a)). Once the parameters of ECONet have been learned, efficient online inference is done by applying it to the whole input CT volume. ECONet is converted to a fully convolutional network for inference (Figure 2 (b)), where appropriate padding is used in the input convolution layer and fully-connected layers are converted to 1x1x1 conv3d (Long et al., 2015). This enables ECONet to efficiently infer a volume with likelihood for each voxel within image $X$.

### 2.3. Scribbles-balanced Cross-Entropy Loss

User-scribbles suffer from class imbalance problem, resulting from the user-interactions being biased towards the object of interest. In addition, during the course of an interactive session, the user may focus on labelling different segments, which results in dynamically changing class imbalance in $S$ (Wang et al., 2016). To address this, we utilize a scribbles-balanced cross-entropy loss (Kukar et al., 1998; Ho and Wookey, 2019), with dynamically changing class weights from scribbles distribution.

Given a model with parameters $\theta$, the foreground likelihood from this model is defined as $p_i = P(s_i = 1|X, \theta)$. Then, the scribbles-balanced cross-entropy loss is:

$$\mathcal{L}(p, y) = -\sum_i \left( w^f y_i \log p_i + w^b (1 - y_i) \log(1 - p_i) \right) \tag{1}$$

where $w^f$ and $w^b$ are scribble-based class weights for foreground and background, respectively, and are defined as: $w^f = |S|/|S^f|$ and $w^b = |S|/|S^b|$.

## 3. Experimental Validation

We compare our proposed ECONet with existing state-of-the-art methods in online likelihood inference, which are Histogram (Boykov and Jolly, 2001), Gaussian Mixture Model (GMM) (Rother et al., 2004) and DybaORF-Haar-Like (Wang et al., 2016). In addition, to show the effectiveness of learning features in ECONet, we define ECONet-Haar-Like that replaces the first convolution layer of ECONet with hand-crafted haar-like features (Jung et al., 2013) and learns the three fully-connected layers. Both DybaORF-Haar-Like and ECONet-Haar-Like utilize our GPU-based implementation of 3d haar-like features, available at: https://github.com/masadcv/PyTorchHaarFeatures. A GPU-based implementation of GMM is used (MONAI Consortium, 2020). DybaORF was implemented using CPU-based Random Forest implementation from (Pedregosa et al., 2011). All experiments were performed on Tesla V100 GPU with 32 GB memory. For user interactions, we utilized the scribbles-based interactive segmentation tools from project MONAI Label[1] (Diaz-Pinto et al., 2022).

**Data.** We use the UESTC-COVID-19 dataset for experimental validation and comparison of ECONet with existing methods (Wang et al., 2020). This dataset contains a total of 120 CT volumes with lung lesion labels, of which 50 are by expert annotators and 70 are by non-expert annotators. In order to compare robustness of our proposed ECONet against expert annotators, we use only the 50 CT volumes labelled by experts for all our experiments. In our validation, the ground-truth labels are only used for generating interactions with a synthetic scribbler and to compute evaluation metrics.

**Training Parameters.** Adam optimizer (Kingma and Ba, 2014) with 200 epochs and an initial learning rate of 0.01 dropped to 0.001 at 140th epoch is used for training of ECONet-based methods. Dropout probability of 0.3 is used during training for all fully-connected layers. The size of each layer in ECONet is selected through line search ablation experiments (see Appendix A), which are as follows: (i) input patch and conv3d kernel size is $7 \times 7 \times 7$ ($K = 7$), (ii) number of filters in input conv3d is 128 and (iii) fully-connected layer sizes are $32 \times 16 \times 2$. The best performing configuration from (Wang et al., 2016) are used for DybaORF, which are 50 trees with maximum tree depth of 20 and minimum samples for split equal to 6. GMM-based method uses 20 Gaussians for each GMM, whereas in the Histogram-based method 128 bins were used to build each histogram. Similar to (Wang et al., 2016), likelihood from ECONet (and all comparison methods) is spatially regularized by applying GraphCut using max-flow/min-cut algorithm (Boykov and Jolly, 2001). Following (Luo et al., 2021), we use $\lambda = 5.0$ and $\sigma = 0.1$ for GraphCut regularization.

---

1. https://github.com/Project-MONAI/MONAILabel

Table 1: Quantitative comparison of online likelihood generation methods using synthetic scribbler from Section 3.1. Mean and standard deviation of DICE (%), ASSD, Time (s) and Synthetic Scribble Voxels Required ($S$) is reported.

| Method | DICE (%) | ASSD | Time (s) | Synthetic Scribbles ($S$) |
|---|---|---|---|---|
| **ECONet (proposed)** | **82.81± 8.77** | **7.57±14.65** | 2.03±1.79 | **2605±2929** |
| **ECONet-Haar-Like** | 71.61±12.43 | 20.28±36.24 | 0.59±0.09 | 3737±2471 |
| **DybaORF-Haar-Like (Wang et al., 2016)** | 66.81±14.92 | 40.81±46.40 | 6.33±1.63 | 11699±7383 |
| **GMM (Rother et al., 2004)** | 50.96± 5.35 | 77.76±38.77 | 0.12±0.06 | 13502±2209 |
| **Histogram (Boykov and Jolly, 2001)** | 49.63± 0.37 | 82.09±31.60 | 0.21±0.06 | 18862±2928 |

**Evaluation Metrics.** Segmentation results from each method are compared against ground truth labels from experts annotators from UESTC-COVID-19 dataset using Dice similarity (DICE) and average symmetric surface distance (ASSD) metrics (see Appendix B for more details) (Luo et al., 2021). Mean and std values for DICE/ASSD are computed by averaging/std a list of per sample metric values. In addition, we also evaluate comparison methods on their online training and inference execution time (Time) as well as the number of voxels with scribbles (S) needed for achieving a given DICE and ASSD score.

### 3.1. Quantitative Comparison using Synthetic Scribbler

We employ a synthetic scribbling method based on the training method in (Wang et al., 2018b). The proposed synthetic scribbler first compares the inferred segmentation label against the ground truth to identify each mis-segmented regions. For the first interaction, where the network is randomly initialized, ground truth is used as mis-segmented region. Following this, each under-segmented (false negative) and over-segmented (false positive) region is localized using connected component analysis. Let $V_m$ define the volume of a given under-segmented or over-segmented region, the synthetic scribbler labels $n$ voxels randomly within that region. $n$ is set to 0 if $V_m < 6^3$ and otherwise to $\lceil V_m/10^3 \rceil$ based on empirical experiments. A likelihood based segmentation label is then inferred using a comparison method with these synthetic scribbles. This synthetic interaction process is repeated 10 times and the metrics corresponding to the final interaction are reported. Note that since the number of synthetically scribbled voxels directly depends on the volume of a given under/over-segmented region, therefore the amount of voxels required by each method directly relate to how well that method performs. An ideal method needs the least amount of synthetic interactions to achieve the best accuracy.

Table 1 shows quantitative comparison of the comparison methods using the proposed synthetic scribbler. It can be observed that ECONet outperforms all existing state-of-the-art in terms of accuracy, while requiring least number of synthetically scribbled voxels. In terms of efficiency, online training and inference of the proposed ECONet takes around 2 seconds combined, which is significantly faster as compared to 6 seconds for DybaORF-Haar-Like, however it is slower than methods that do not learn a classifier (i.e., GMM and Histogram).

To further analyze the quantitative results, we visualize the percentage of dataset samples below a given DICE score for all methods in Figure 3 (a). It can be observed that 70% of the dataset achieves above 80% DICE using ECONet. As compared to this, ECONet-Haar-Like has 50% and DybaORF-Haar-Like has 15% samples above 80% DICE. It can

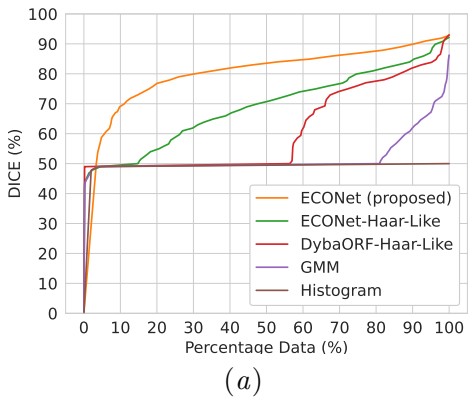
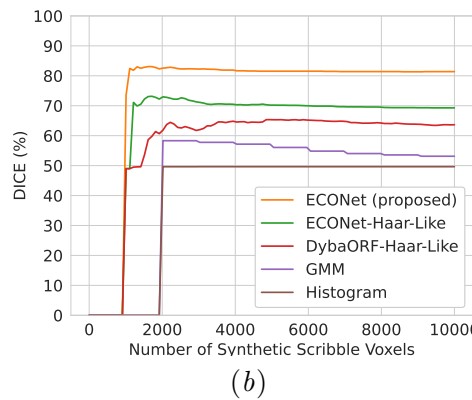

$$(a) \qquad\qquad\qquad\qquad (b)$$

Figure 3: Quantitative analysis using synthetic scribbler on UESTC-COVID-19 dataset, shows (a) percentage of dataset samples that are below a given DICE score, and (b) number of synthetically scribbled voxels ($S$) needed to achieve corresponding DICE score for method. Plateaus in (b) indicate that a method does not require further interactively labelled voxels to improve accuracy.

also be observed that both GMM and Histogram method failed in most cases achieving 50% DICE, which indicates labelling of all voxels with same label.

Figure 3 (b) presents analysis of the amount of synthetic scribbled voxels required to achieve a given DICE for all comparison methods. It can be observed that for ECONet, an average of ∼1600 labelled voxels achieve DICE > 80%. Similarly, ECONet-Haar-Like requires ∼1650 labelled voxels to achieve DICE > 70%. Unlike ECONet-based methods, DybaORF-Haar-Like requires significantly greater number of labelled voxels (∼5000) and only achieves DICE > 65%. Both GMM and Histogram fail, with additional labelled voxels having no effect on Histogram. Interestingly, for GMM increasing the number of labelled voxel adversely affects the accuracy resulting in drop in DICE. We believe this is due to the limited representation capability of GMM learning from voxel intensity alone, which is insufficient to model the additional ambiguous variations.

### 3.2. Qualitative Comparison using Scribbles from Non-expert Annotator

A non-expert annotator provided scribble-based interaction for labelling CT volumes from UESTC-COVID-19 dataset. The provided scribbles were used for annotation using learned likelihood methods i.e., ECONet, ECONet-Haar-Like and DybaORF-Haar-Like. Figure 4 shows the qualitative results from this experiment. As can be observed, ECONet is able to provide segmentation labels close to the ground truth, which is due to the learned features that enable the network to better differentiate lung lesions from the background. In our future work, we will do a quantitative analysis to measure the quality of these annotations.

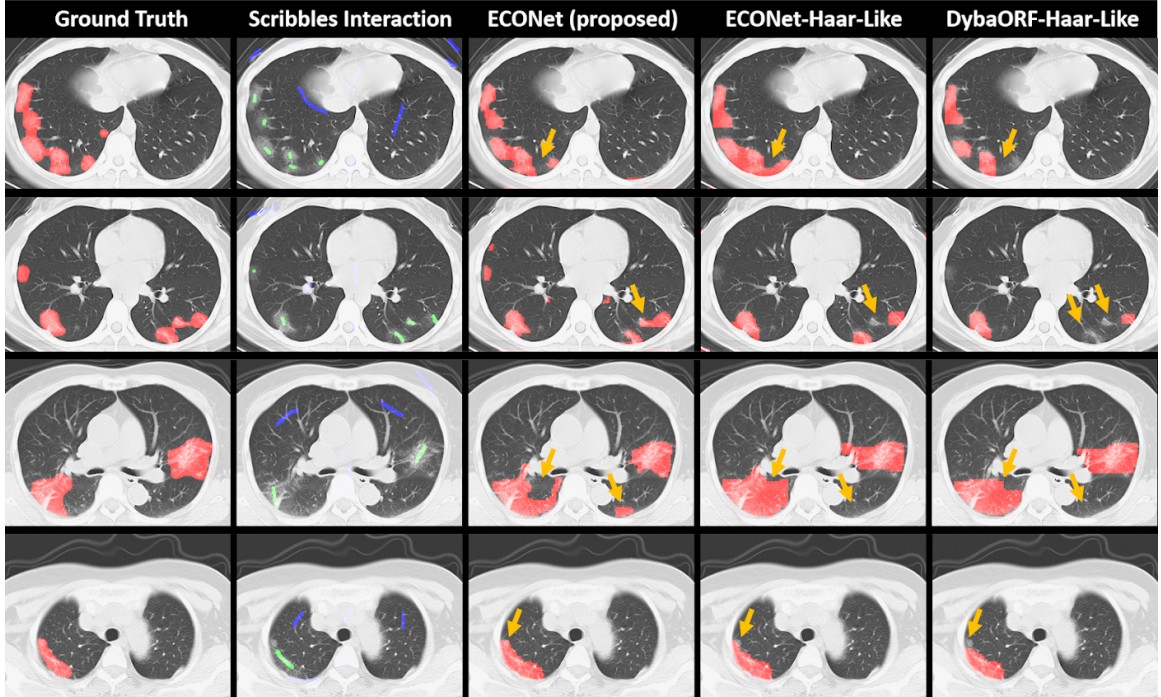

Figure 4: Qualitative comparison of online likelihood methods using scribbles from a non-expert annotator. Segmentations are in red, while foreground and background scribbles are in green and blue, respectively. ↑ shows mis-segmented regions.

## 4. Conclusion and Future Work

We proposed Efficient Convolutional Online Likelihood Network (ECONet) for scribble-based interactive segmentation of lungs lesions in CT volumes from COVID-19 patients. The lightweight architecture of ECONet enabled online training and inference using scribble-based annotations. ECONet was learned online, without the need for any pre-training, from interactive labels for a given CT volume. A method for efficient online learning of ECONet was proposed, which consisted of extracting and using only the patches with user-provided scribble labels. For inference, the network was applied to full volume using a fully convolutional approach. Experimental validation showed that the proposed ECONet outperformed existing state-of-the-art for online likelihood learning on the task of labelling COVID-19 lung lesions.All ECONet-based methods outperformed state-of-the-art DybaORF-Haar-Like method in terms of accuracy as well as online learning efficiency. ECONet achieved 16% higher DICE score in $3\times$ lesser time while requiring around 9000 lesser scribble labelled voxels than DybaORF-Haar-Like.

In our future work, we will use ECONet within interactive segmentation pipelines, where it will enable quick online adaption based on user interactions. In addition, we will study the quality of annotations achieved using ECONet and extend ECONet for multi-class online likelihood based annotation problems.

## Acknowledgments

This project has received funding from the European Union's Horizon 2020 research and innovation programme under grant agreement No 101016131 (icovid project). This project has received funding from the European Union's Horizon 2020 research and innovation program under the Marie Skłodowska-Curie grant agreement TRABIT No 765148. This work was also supported by core and project funding from the Wellcome/EPSRC [WT203148/Z/16/Z; NS/A000049/1; WT101957; NS/A000027/1]. TV is supported by a Medtronic / Royal Academy of Engineering Research Chair [RCSRF1819\7\34].

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

## Appendix A. Experiments for Searching Optimal Layer Sizes for ECONet

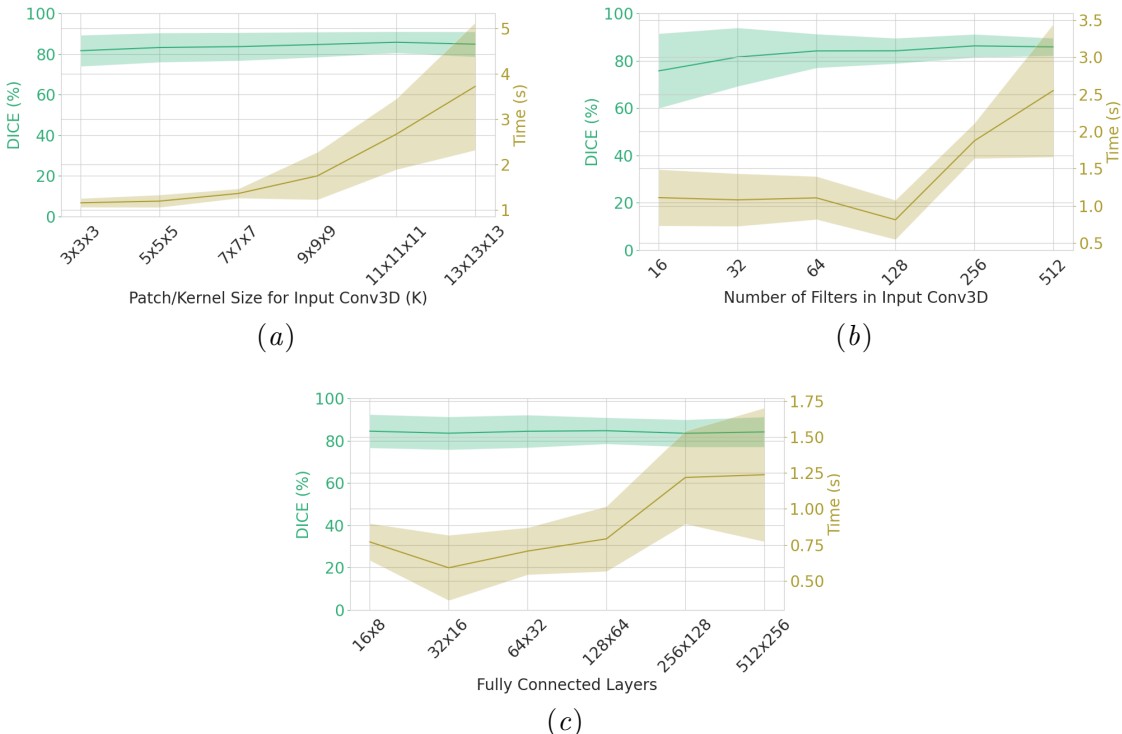

Figure 5: Searching for the optimal layer sizes for ECONet. Shows ablation experiments with varying (a) input patch size/input conv3d kernel size ($K$), (b) number of filters in input conv3d, and (c) size of the fully-connected layers against accuracy DICE (%) and Time (s) for both online inference and training using the corresponding ECONet model. These experiments were performed using 10 randomly choosen CT volumes from UESTC-COVID-19 dataset. Optimal sizes selected using these experiments are: input patch size/input conv3d kernel size $7 \times 7 \times 7$ ($K = 7$) with 128 filters and $32 \times 16$ fully-connected layers. The largest impact on DICE comes from number of filters in (b), which directly corresponds to our observation on requirement of learned features. Increasing different layer sizes in ECONet significantly increases the online training and inference time as evident in these experiments. All ablation experiments are performed on a single Tesla V100 GPU with 32GB memory.

Table 2: Execution time showing online training and online inference time for deeper ECONet models ($\mathbf{L_{num}} > 1$. $\mathbf{L_{num}} = 1$ is the proposed ECONet model.

| Execution time | $\mathbf{L_{num}} = 1$ (proposed) | $\mathbf{L_{num}} = 2$ | $\mathbf{L_{num}} = 3$ | $\mathbf{L_{num}} = 4$ |
|---|---|---|---|---|
| Training 2605 patches, 200 epochs | 2.35 seconds | 177.46 seconds | 770.84 seconds | 1367.09 seconds |
| Inference on $185 \times 127 \times 68$ input | 0.0013 seconds | 0.0018 seconds | 0.249 seconds | 1.02 seconds |

## Appendix B. Computing DICE and ASSD metrics

Given a region $L_{pred}$ segmented using a likelihood inference method and the corresponding ground truth region $L_{gt}$, DICE is defined as:

$$\text{DICE} = \frac{2 \cdot |L_{pred} \cap L_{gt}|}{|L_{pred}| + |L_{gt}|}. \tag{2}$$

ASSD is defined by comparing surface points from a comparison method $T_{pred}$ against surface points $T_{gt}$ from ground truth segmentation label as:

$$\text{ASSD} = \frac{1}{|\,T_{pred}\,| + |\,T_{gt}\,|} \left( \sum_{j \in T_{pred}} d(j, T_{gt}) + \sum_{j \in T_{gt}} d(j, T_{pred}) \right) \tag{3}$$

where $d(i, T_{gt})$ is the shortest Euclidean distance between point $j$ and surface $T_{gt}$.

## Appendix C. Efficiency of Deeper ECONet models

We designed ECONet to be lightweight, while still being adaptable and efficient for online learning. Therefore, we choose one convolutional and 3 fully-connected layers. The size of each layer was chosen by experiments in Appendix A. We note that additional convolutional layers (i.e., deeper network) may improve accuracy, however it will require larger training scribbles data and more epochs, making the method lose its quick adaptability. In addition, as shown in Table 2, the additional layers adversely impact the efficiency of training and inference from ECONet. In this table, we use ECONet configuration reported in paper, i.e. (i) input patch and conv3d kernel size is $7 \times 7 \times 7$ ($K = 7$), (ii) number of filters in input conv3d is 128 and (iii) fully-connected layer sizes are $32 \times 16 \times 2$. For each method with $\mathbf{L_{num}} > 1$, $\mathbf{L_{num}}$ conv3d layers ($K = 7$, num filters = 128) replace the input conv3d. For training, we used 2605 patches (average $S$ in Table 1) and ran 200 epochs following our setting in paper. For inference, we use $185 \times 127 \times 68$ size volume which is the average size of input from UESTC-COVID-19 dataset. We note that both training and inference time were significantly increased with deeper models ($\mathbf{L_{num}} > 1$). This makes deeper models prohibitive for use in an online interactive segmentation setting. All experiments were performed using a single Tesla V100 GPU with 32 GB memory.

