# OpenReview forum: "ECONet: Efficient Convolutional Online Likelihood Network for Scribble-based Interactive Segmentation"
_MIDL.io/2022/Conference — MIDL 2022_

### Official Review · Reviewer_TLdU · 2022-01-04

**Confidence:** 5
**Preliminary Rating:** 4
**Recommendation:** Oral

**Summary:**

This paper proposed a convolutional neural network, called ECONet, that can be learned online while the annotator provides scribble-based interaction. To accelerate learning from only the samples labeled through user interactions, a patch-based approach is used for training the network, with a weighted cross-entropy loss used to address the class imbalance issue that may result from user interactions. During online inference, the learned network is applied to the whole input volume using a fully convolutional approach. Experiments were performed on the task of annotating lung lesions associated with COVID-19, based on 120 CT volumes with lung lesion labels.

**Strengths:**

(1) The proposed method is a distinctive likelihood model in an online and data-light manner using only scribbles-based labels.

(2) The proposed network is lightweight, with a single convolutional layer and three fully-connected layers for feature extraction and classification.

(3) A weighted cross-entropy loss is used to address the class imbalance issue.

(4) Experimental evaluation is extensive


**Weaknesses:**

(1) The influence of the patch size K is unclear.

(2) The influence of annotation quality of Scribbles S was not quantitively evaluated.

(3) The network only contains 1 convolutional layer and 3 FC layers, while it is not clear whether using a deeper network (e.g., 2 or 3 or 4 convolutional layers) helps.


**Deanonymize Review:**

no

**Detailed Comments:**

(1) The influence of the patch size K is unclear.
It was mentioned that “Scribbles S provided by an annotator at a given stage only label a small subset of voxels within a given image volume X. Based on this observation, we minimize the computational budget required to perform training passes on ECONet by extracting and learning only from patches with K×K×K dimensions, each centered around a voxel with user-scribble”. This hyperparameter is essential for the final performance. Thus, it would be interesting to explicitly investigate its influence.


(2) The influence of annotation quality of Scribbles S was not quantitively evaluated.
It is good to ask a non-expert annotator to provide scribble-based interaction for labeling CT volume, as mentioned in Section 3.2. It could be better to further investigate the annotation quality of this annotator, and also study the influence of the quality of such scribble-based interactions for the segmentation performance.


(3) The network only contains 1 convolutional layer and 3 FC layers. This will make it lightweight and adapt to online learning settings. But it is interesting to study whether using a deeper network (e.g., 2 or 3 or 4 convolutional layers) would help boost the performance, and how much computation burden or time complexity will be introduced.


**Paper Type:**

both

**Questions To Address In The Rebuttal:**

(1) The influence of the patch size K is unclear.

(2) The influence of annotation quality of Scribbles S was not quantitively evaluated.

(3) The network only contains 1 convolutional layer and 3 FC layers, while it is not clear whether using a deeper network helps.


**Special Issue:**

yes

---

### Official Review · Reviewer_kaUZ · 2022-01-22

**Confidence:** 3
**Preliminary Rating:** 5
**Recommendation:** Oral, Poster

**Summary:**

This paper presents a novel approach for deep learning-based interactive segmentation.Input to the network is provided by means of "scribbles". The method does not require pre-training and is lightweight. It permits the labelling of the entire volume at inference. The method has been tested on a set of COVID-19 images and compared to existing methods.

**Strengths:**

This is an interesting and well-written paper. Coverage of the literature is good and it presents a relatively simple but efficient method. The  architecture consists in a fully convolutional layer for feature learning and fully connected layers for classification. The main idea to speed things up is to limit the training to patches. A good comparative study with existing methods is perfomed and suggests that the proposed methods has an edge over existiing ones, at least for the application on wihch it has been tested. Overall, this seems to be an approach that will have practical utility.

**Weaknesses:**

No real weaknes but a few points of clarification

1) What is the value of K and what is the sensitivity of the results on this value. Do the authors expect that this value will need to be adapted for various applications.

2) I am not really clear on how the experiements are done. Apparently, results have been generated with 50 annotated volumes. Is scribble-based training started from scratch for each volume?

3) How are the DICE and ASSD values computed. Per object and then averaged?

**Deanonymize Review:**

no

**Final Rating After The Rebuttal:**

5: Strong Accept

**Justification Of The Final Rating:**

I only had  a few questions/comments and they have been successfully addressed by the authors. The paper presents an interesting  approach that will likely be of value to the community. The validation study is solid and the code available.

**Paper Type:**

both

**Questions To Address In The Rebuttal:**

Please address the questions asked above and the issue of cross-application robustness. Here, parameters  have been set using ablation experiments. Do the authors anticipate that such experiments will need to be repeated for other applications, e.g., segmentation of brain tumors.

Will the code be made available?


**Special Issue:**

no

---

### Official Review · Reviewer_or3i · 2022-01-25

**Confidence:** 3
**Preliminary Rating:** 4
**Recommendation:** Oral, Poster

**Summary:**

The authors propose an efficient way to perform segmentation using online learning with scribble-based interactions. Their proposed method, called ECONet, is used to segment lung lesions from CT volumes of COVID-19 affected patients. They tackle the issues of limited expert-annotated labels of CT volumes for lung lesions and a large overlap of background and foreground (lesion) voxels due to visual similarity. The authors also implement a weighted cross-entropy loss to address the class imbalance problem. The results presented show that ECONet outperforms existing SOTA while reducing execution time and the number of scribbled voxels needed.

**Strengths:**

- The paper addresses some major challenges presented by medical data in developing well-performing segmentation models
- ECONet presents an efficient way for segmentation using online learning. Although the authors present one use-case, I can see this method being applied for other tasks as well
- The results provided look promising; the authors provide both quantitative and qualitative outcomes.
- The paper is well presented and is easy to follow with substantial literature review

**Weaknesses:**

- There are no baselines presented. The proposed method is changed slightly to use hand-engineered features but I would like to see if any other methods were tried. How do offline methods perform?
- Code is not available during the review although the authors do plan to release it after acceptance
- The size of the S (subset of voxels) has not been mentioned which would play an important role in performance


**Deanonymize Review:**

no

**Detailed Comments:**

1. In the abstract, the authors should mention that the results presented are from synthetic scribbles
2. The contribution section does not specify the SOTA methods against which the performance has been calculated
3. In 2.3, was empirical evidence used for choosing the weights $w^f$ and $w^b$? Did you try other methods for mitigating class imbalance, like focal loss or some form of domain adaptation?
3. In the data section, what subset of voxels within the CT volume were labeled by the expert annotators. Also, it is mentioned that 50 out of the 120 volumes had labels, what about the rest of the volumes? How were they utilized?
4. In 3.1, how do you identify and localize under/over-segmented regions?
5. Was a validation set used for hyperparameter tuning?

**Final Rating After The Rebuttal:**

5: Strong Accept

**Justification Of The Final Rating:**

I thank the authors for the detailed comments. My main questions were answered satisfactorily and some of the considerations were also added in the revised version. This paper presents an interesting approach to online learning for medical applications, and I can see it being applied to other tasks as well.

**Paper Type:**

methodological development

**Questions To Address In The Rebuttal:**

The paper presents an interesting idea for a task which is relevant in the medical domain. Main points to be addressed are in the weaknesses and questions about certain parts of the paper in the detailed comments.

**Special Issue:**

no

---

### Meta-Review · Area_Chair_T6xV · 2022-02-21

**Recommendation:** Accept (Poster)
**Confidence:** 5

**Metareview:**

The authors have carefully addressed reviewers' critiques. I recommend that the paper be presented as a poster because this work presents an "interactive" solution to image segmentation, and a poster presentation gives the authors an opportunity to demonstrate their system extensively and engage in deep discussions with the MIDL audience.

---

### Decision · Program_Chairs · 2022-02-28

Accept